# The Impact of *Candida albicans* in the Development, Kinetics, Structure, and Cell Viability of Biofilms on Implant Surfaces—An In Vitro Study with a Validated Multispecies Biofilm Model

**DOI:** 10.3390/ijms25063277

**Published:** 2024-03-14

**Authors:** Enrique Bravo, Marion Arce, Honorato Ribeiro-Vidal, David Herrera, Mariano Sanz

**Affiliations:** 1ETEP (Etiology and Therapy of Periodontal and Peri-Implant Diseases) Research Group, Faculty of Dentistry, Complutense University, 28040 Madrid, Spain; ebravofe@ucm.es (E.B.); davidher@odon.ucm.es (D.H.); 2Department of Conservative Dentistry, Faculty of Dentistry, University of Chile, Santiago 8380544, Chile; marce05@ucm.es; 3Laboratory of Oral Microbiology and Immunology, Faculty of Dentistry, University of Chile, Santiago 8380544, Chile; 4Department of Periodontology, Faculty of Dentistry, University of Porto, 4200-393 Porto, Portugal; honorato.vidal@gmail.com

**Keywords:** peri-implantitis, oral biofilm, *Candida albicans*, scanning electron microscopy, confocal laser scanning microscopy, quantitative polymerase chain reaction

## Abstract

This study aimed to evaluate the impact of *Candida albicans* on subgingival biofilm formation on dental implant surfaces. Scanning electron microscopy (SEM) and confocal laser scanning microscopy (CLSM) were used to compare biofilm structure and microbial biomass in the presence and absence of the fungus after periods of 24, 48, and 72 h. Quantitative polymerase chain reaction (qPCR) was used to quantify the number of viable and total micro-organisms for each of the biofilm-forming strains. A general linear model was applied to compare CLSM and qPCR results between the control and test conditions. The biofilm developed with *C. albicans* at 72 h had a higher bacterial biomass and a significantly higher cell viability (*p* < 0.05). After both 48 and 72 h of incubation, in the presence of *C. albicans*, there was a significant increase in counts of *Fusobacterium nucleatum* and *Porphyromonas gingivalis* and in the cell viability of *Streptococcus oralis*, *Aggregatibacter actinomycetemcomitans*, *F. nucleatum*, and *P. gingivalis*. Using a dynamic in vitro multispecies biofilm model, *C. albicans* exacerbated the development of the biofilm grown on dental implant surfaces, significantly increasing the number and cell viability of periodontal bacteria.

## 1. Introduction

The use of dental implants is currently the most widespread strategy for the rehabilitation of total or partial edentulism, resulting in long-term satisfactory success rates [1,2,3]. However, these implant-supported restorations are susceptible to complications during their function, both mechanical and biological. At the last World Workshop on the Classification of Periodontal and Peri-implant Diseases and Conditions (2017), peri-implant diseases were classified as peri-implant mucositis and peri-implantitis [4], with estimated prevalence ranges between 43% and 47% for peri-implant mucositis and 20% and 22% for peri-implantitis [5]. The primary etiological factor of peri-implantitis is the biofilm formed on dental implants and their restorative component surfaces, triggering a chronic inflammatory response, eventually leading to bone destruction and progressive loss of implant osseointegration [4,6,7,8,9]. Submarginal biofilms are structurally and functionally organized complex microbial communities, consisting mainly of bacteria but also viruses, protozoa, and, to a greater extent, fungi, that synthesize an extracellular polymeric matrix, which binds cells together and anchors them to biotic or abiotic surfaces susceptible to colonization [10,11,12,13,14]. The development of dysbiotic biofilms, with changes in the relative proportions of the bacterial communities, has been associated with the etiology and pathogenesis of periodontal and peri-implant diseases and the decreasing efficacy of antimicrobial treatments [15].

The most frequent fungal pathogen in the oral cavity is *Candida albicans*, a dimorphic facultative anaerobic fungus that is usually present as a yeast under favorable environments, although it usually presents as a filamentous fungus under unfavorable conditions [16]. These differential conditions are related to nutrient availability, environmental atmospheric composition, or the presence of antifungal agents [17].

The presence of *C. albicans* in subgingival pockets has been reported at higher rates in subjects with periodontitis compared with periodontally healthy individuals [18], although there is a high heterogeneity in the reported prevalence of *C. albicans* in the subgingival microbiota of periodontitis patients, ranging from 14.6% to 87.5% [19,20]. Similarly, in peri-implantitis, the presence of *C. albicans* is higher than around implants with healthy peri-implant tissues or in peri-implant mucositis sites [21]. In different studies, *C. albicans* was detected in 27%, 15.8%, 77.6%, and 76.2% of patients with peri-implantitis, versus 0%, 10%, 12.2%, and 9.8% in patients with peri-implant health [7,22,23,24]. This high variability could be due to the different identification methods used in the different studies.

The pathogenicity of *C. albicans* is mediated by its adhesion to the implant surface in a process favored by salivary mucin and albumin [25]. Once adhered, its growth generates hyphae and secretes hydrolytic enzymes, basically proteases, lipases, and hemolysins, which may activate the inflammatory response of the host soft tissues [26,27,28]. The inflammatory response to *C. albicans*-infected epithelial cells is mediated by proinflammatory cytokines, which further contribute to the chronic inflammatory response characteristic of peri-implant disease lesions and, eventually, to the tissue destruction and alveolar bone resorption characteristic of peri-implantitis [29,30,31].

Validated biofilm models have been used to study the interactions between micro-organisms and test in vitro antimicrobial therapies [32,33,34,35]. The pathogenic mechanisms of the potential interactions between *C. albicans* and the bacteria present in subgingival/submarginal biofilms have been mostly studied in vitro, using culturing or static biofilm models [36,37,38]. These studies have demonstrated that *C. albicans* influences biofilm architecture and favors the presence and virulence of certain periodontal pathogenic bacterial strains [39,40,41]. However, the overall effect of *C. albicans* on a multispecies biofilm or on biofilm formation on dental implant surfaces has not yet been studied. It was, therefore, the aim of this in vitro study to evaluate the impact *of C. albicans* on the development, kinetics, structure, and viability of biofilm formation on dental implant surfaces in a validated multispecies dynamic model. Further knowledge about these interactions may help in the development of new therapies aimed at the control of periodontal and peri-implant diseases.

## 2. Results

### 2.1. Scanning Electron Microscopy (SEM) Analysis

Figure 1 depicts the SEM images of biofilms grown on the implant surfaces after 24, 48, and 72 h in the presence (test) and absence (control) of the fungus *Candida albicans*, clearly showing a differential biofilm morphology.

After 24 h of incubation, *C. albicans* presented as yeast (Figure 1G) and did not affect the structure of the biofilm, composed at this first stage by cocci (*Streptococcus oralis* and *Veillonella parvula*), and rods and spindle-shaped bacteria, corresponding to *Actinomyces naeslundii* and *Fusobacterium nucleatum* (Figure 1A,D).

At 48 h, *C. albicans* started to manifest as filaments, forming pseudohyphae to which *F. nucleatum* and cocci bacteria were anchored (Figure 1H). Compared to the 24-h biofilm, there was a higher density of spindle-shaped bacteria and cocci, corresponding to *F. nucleatum* and *Aggregatibacter actinomycetemcomitans*, respectively. In the test biofilms (with *C. albicans*) (Figure 2E), the presence of coccobacillary forms of *Porphyromonas gingivalis* was more frequent than in the control biofilms (Figure 2B).

At 72 h, cellular aggregates formed by spindles, cocci, and coccobacilli (*F. nucleatum*, *A. actinomycetemcomitans* and *P. gingivalis*, respectively) were deposited on the pseudohyphae of *C. albicans*, presenting as a mixed mature biofilm with a higher biomass than in the previous time intervals (Figure 1I). The test biofilms demonstrated a higher bacterial density, as shown in Figure 1F,C. In fact, control biofilms at 72 h presented a lower bacterial density than the test biofilms at 48 h. 

### 2.2. Confocal Laser Scanning Microscopy (CLSM) Analysis

*C. albicans* increased its size on the implant surface as the biofilm was maturing after 24, 48, and 72 h of incubation (Figure 2 and Figure 3). Figure 2A–F depict CLSM images representative of control and test biofilms, respectively, clearly showing the impact of *C. albicans* on the biomass of the test biofilms. Figure 2G-I show *C. albicans* in the test biofilms. The kinetics of the development of both biofilms are shown in Figure 3.

After the first 24 h, the bacterial biomass of biofilms grown in the absence of *C. albicans* (7.21 µm^3^/µm^2^ (standard deviation, SD = 4.50)) and in the presence of the fungus (6.63 µm^3^/µm^2^ (SD = 4.63)) showed no differences (Figure 2A,D). The viability percentages, 60.20% (SD = 20.19%) and 63.37% (SD = 9.70%), respectively, were also similar (Figure 3). The biomass of *C. albicans* incorporated into the test biofilm was 5.95 µm^3^/µm^2^ (SD = 3.18) (Figure 2G).

Biofilms formed after 48 h also did not show differences between test and control biofilms. Control biofilms had a bacterial biomass of 10.02 µm^3^/µm^2^ (SD = 6.21) and test ones of 9.69 µm^3^/µm^2^ (SD = 2.14) (Figure 2B,E). There were also no differences between the cell viability of the two biofilms, 52.47% (SD = 5.00%) for control biofilms and 58.50% (SD = 18.07%) for test biofilms (Figure 3). *C. albicans* had a biomass of 8.86 µm^3^/µm^2^ (SD = 2.78) (Figure 2H).

In contrast, the mature biofilms developed in the presence of *C. albicans* after 72 h of incubation showed higher bacterial biomass than control biofilms (12.88 µm^3^/µm^2^ (SD = 8.77) and 8.31 µm^3^/µm^2^ (SD = 4.70, respectively)). Furthermore, as seen in Figure 2C,F, and 3, the bacterial viability of the biofilms developed without the fungus was 31.34% (SD = 9.81%), while in the mixed biofilm, it was 66.70% (SD = 10.05%), a difference that was statistically significant (*p* < 0.05). Figure 2I shows the biomass corresponding to *C. albicans*, which was 18.91 µm^3^/µm^2^ (SD = 6.31). 

### 2.3. Quantitative Polymerase Chain Reaction (qPCR) Analysis 

Figure 4 shows counts of total and viable cells in biofilms developed at 24, 48, and 72 h, expressed as colony-forming units (CFU)/mL for each bacterial species and *C. albicans* in test biofilms, together with cell viability and the percentage of live cells to total counts.

At 24 h of incubation, the growth and viability of *C. albicans* were limited. There were no statistically significant differences in counts for any of the six biofilm-forming bacterial species when comparing test and control biofilms.

At 48 h, the development of *C. albicans* increased, and in the test biofilms, the counts and viability of *F. nucleatum* and *P. gingivalis* were significantly higher. The same pattern was observed for *A. naeslundii*. Cell viability of *A. actinomycetemcomitans* was also significantly higher in test biofilms.

At 72 h, a similar pattern occurred, with higher growth of *C. albicans*, when compared to previous intervals, and larger counts of *F. nucleatum*, *P. gingivalis,* and *A. naeslundii* in test biofilms when compared to control biofilms. At this stage, also the number of viable cells of *S. oralis*, *F. nucleatum*, *P. gingivalis,* and *A. actinomycetemcomitans* were significantly higher compared to the controls. 

## 3. Discussion

In the present study, a validated in vitro multispecies dynamic biofilm model was used to assess the influence of *C. albicans* on biofilms developed on dental implant surfaces. The selection of species for the biofilm model was based on selecting a representative sample of the diversity of the subgingival biofilm including early, intermediate, and late colonizers. This selection included gram-positive and gram-negative bacterial strains as well as bacteria of different nutritional and environmental requirements. Bacterial counts determined by qPCR analysis indicated that, after 48 and 72 h of growth, the number and cell viability of *F. nucleatum* and *P. gingivalis* were significantly higher in biofilms developed in the presence of the fungus. Similarly, the proportion of live cells of *A. actinomycetemcomitans* and *S. oralis* also increased significantly in the presence of *C. albicans* in mature biofilms (72 h) (Figure 4 and Appendix A). Similar results were obtained by CLSM analyses, depicting a significantly higher overall size and cell viability in mature biofilms (72 h) in the presence of *C. albicans* (Figure 2 and Figure 3). 

The progressive filamentation of *C. albicans* cells observed in the SEM analysis (Figure 1) may have been favored by the experimental conditions of the biofilm model used, since they simulate oral cavity conditions (pH 7, 37 °C, anaerobic environment) and the presence of gram-negative bacteria (*V. parvula*, *F. nucleatum*, *P. gingivalis,* and *A. actinomycetemcomitans*) [17]. The filamentation process is mediated by the *ROB1^946S^* allele [42]. In the maturation of *C. albicans*-associated biofilms, the filamentation process led to the attachment to implant surfaces of hyphae and yeast-like (sessile) cells, associated with microcolonies of rods and spindle-shaped bacteria, embedded in an extracellular matrix. This morphology coincides with other previous descriptions of *C. albicans*-associated biofilms [43]. The impact of *C. albicans* on biofilm formation shown in the present study, demonstrating significantly higher biofilm biomass and higher percentages/counts of total and viable bacterial strains, may be exacerbated by the demonstrated activation in the expression of hydrolytic enzymes by *C. albicans* in the presence of periodontal bacteria, which may further compromise the host immune response and enhance the resistance of the resulting biofilm to antifungal agents [36,44].

One of the possible relevant findings of the present study is the specific impact of *C. albicans* on the percentage of viable cells in *P. gingivalis* (Figure 4 and Appendix A). This effect may be due to the enhanced anaerobic environment generated by the fungal hyphae due to oxygen consumption [45], clearly depicted within the biofilm architecture shown by SEM. Additionally, Interlin InlJ has been involved in the expression of *P. gingivalis* genes responsible for the interaction with *C. albicans* hyphae [46]. It has also been reported that adhesins Als3 and the proteases Sap6 and Sap9 of *C. albicans*, together with the gingipains of *P. gingivalis,* may favor the invasion of these micro-organisms in epithelial cells and fibroblasts [39,47]. Along the same lines, citrullination, mediated by peptidyl arginine deiminase (PPAD) secreted by *P. gingivalis*, may favor the adhesion of this bacterial species to the cell wall of *C. albicans*, thus increasing its viability under aerobic conditions [48]. The competition for iron sources that may occur between *P. gingivalis* and *C. albicans* under conditions such as those of the present study, where this nutrient is limited, may also explain the increased viability of the bacteria in the mixed biofilm. Furthermore, this competition may also favor the resistance of *P. gingivalis* to antimicrobial substances by increasing the expression of virulence genes [49]. Thus, the beneficial effect of *C. albicans* on *P. gingivalis* could increase the pathogenic capacity of this periodontal pathogen. In contrast to the results from the present investigation, Cavalcanti et al. (2016) reported that *P. gingivalis* exerted an opposite influence on *C. albicans* by inhibiting its hyphal production. In the model used in the present investigation, the concomitant presence of *Streptococcus* and *Actinomyces* species may have reverted to this inhibition [36,37]. In fact, other authors have argued that the effect of this interaction is dependent on the fungal strain, the composition of the medium, and the streptococcal population present [50].

*C. albicans* also significantly increased the vitality of *F. nucleatum* in the multispecies biofilm (Figure 4 and Appendix A). This could be due to the interaction between the bacterial adhesin radD and the fungal cell wall mannoprotein FLO9, thus facilitating a specific dual aggregation and enhanced growth of *F. nucleatum* [51,52]. This increased growth may enhance the bridging role of *F. nucleatum* between primary colonizers and the late colonizers *P. gingivalis* and *A. actinomycetemcomitans*, an effect that has already been attributed to *C. albicans* [41]. Conversely, another in vitro study indicated that *F. nucleatum* could inhibit the filamentation process of *C. albicans* by limiting its ability to kill macrophages and, thus, attenuating its pathogenic potential [53]. Similarly, the presence of *A. actinomycetemcomitans* through its autoinducer Quorum Sensing-2 (AI-2) molecule inhibits fungal hyphal formation and *C. albicans* aggregation [38]. However, the quantitative results from the present study indicated that *C. albicans* increased the survival rate of *A. actinomycetemcomitans* in mature biofilms. This phenomenon may suggest that the protective anaerobic environment generated by the hyphae and the consequent increased development of *F. nucleatum* spindles would favor the survival of *A. actinomycetemcomitans* in mature biofilms. Further studies are needed to elucidate this specific dual interaction. 

The increase in live cells of the initial colonizer, *S. oralis,* in the mature biofilm was also favored by *C. albicans*. This effect can be explained by the binding of the cocci to the gtfR glucan-binding domain, the main component of the cell wall of *C. albicans* [54]. *S. oralis* is also supposed to induce filamentation of *C. albicans*, which may enhance the invasiveness of fungal and bacterial cells into host epithelial cells [55,56].

Based on the above interactions, Figure 5 shows a comparison of biofilms developed in the presence and absence of *C. albicans*. The presence of the fungus stimulates a more robust and compact mature biofilm, where anaerobic environments are enhanced, which may stimulate the proliferation and growth of more pathogenic bacteria.

Consistent with the results from the present investigation, in vivo studies have also reported that *C. albicans* may exert an important pathogenic effect in the later stages of peri-implantitis, when the biofilm is already established [36]. In fact, case-control studies have demonstrated a higher presence of *C. albicans* in the peri-implant sulcus of patients with peri-implantitis compared with those with healthy peri-implant tissues [57]. Similarly, the presence of hyphae in connective tissue specimens of peri-implantitis has been demonstrated in association with *P. gingivalis*, *A. actinomycetemcomitams* and *P. intermedia* [58], as well as with *V. parvula*, *Tannerella forsythia* and *Parvimonas micra* [59]. A deeper understanding of the interactions of *C. albicans* with the virulence of the different individual bacterial species within the subgingival/submarginal biofilms may help to better understand its pathogenicity and its resistance to antimicrobial strategies. For example, the β,1-3 glucan in the cell wall of *C. albicans* has been shown to modulate the tolerance of periodontal bacterial anaerobes to different antibiotics [60]. This knowledge may also help to design more effective preventive strategies, such as those based on the use of pre- or probiotics [37] or agents aimed at preventing this dysbiotic effect.

The experimental procedures used for the development of the present study are not free of limitations that should be acknowledged. First, although the biofilm model attempts to mimic the conditions of the oral cavity, there are specific individual variables that cannot be reproduced. In addition, natural subgingival/submarginal biofilms may be composed of hundreds of species, whereas the model used is composed of six bacterial species that are intended to be representative of different types of colonizers. Finally, the accuracy of the data obtained is limited due to the high experimental variability linked to in vitro work with live micro-organisms. 

Considering the acknowledged limitations, the statistical evaluation of the obtained results allows us to conclude that *C. albicans* has a significant impact on the growth, dynamics, structure, and viability of subgingival/submarginal biofilms formed on implant surfaces, favoring an increase in the development of *P. gingivalis*, *F. nucleatum*, *A. actinomycetemcomitans,* and *S. oralis*. In conclusion, the effect on the biofilm and on the periodontal pathogens *P. gingivalis* and *A. actinomycetemcomitans* exerted by *C. albicans* may impact the initiation and progression of periodontal and peri-implant diseases.

## 4. Materials and Methods

### 4.1. Microbial Strains and Culture Conditions

Bacterial strains Streptococcus oralis CECT 907T, *Actinomyces naeslundii* ATCC 19039, *Veillonella parvula* NCTC 11810, *Fusobacterium nucleatum* DMSZ 20482, *Porphyromonas gingivalis* ATCC 33277 and *Aggregatibacter actinomycetemcomitans* DSMZ 8324 were used. They were grown on blood agar plates (Blood Agar Oxoid No 2; Oxoid, Basingstoke, UK), supplemented with 5% (*v*/*v*) sterile horse blood (Oxoid, Basingstoke, UK), 5.0 mg/L haemin (Sigma, St. Louis, MO, USA), and 1.0 mg/L menadione (Merck, Darmstadt, Germany) at 37 °C for 24–72 h under anaerobic conditions (10% H_2_, 10% CO_2_, and N_2_ balance). The fungal strain *Candida albicans* SC 5314 was grown on yeast-peptone-glucose (YPD) agar plates (2% glucose (Panreac, Barcelona, Spain), 2% peptone (Life Technologies, Detroit, MI, USA), 1% peptone yeast extract (Life Technologies, Detroit, MI, USA), and 2% agar (Becton, Dickinson and Company, Sparks, MD, USA)) at 37 °C for 24 h under aerobic conditions. 

Pure cultures of each strain were grown for 24 h under anaerobic conditions in protein-enriched brain heart infusion (BHI) medium (Becton, Dickinson and Company, Franklin Lakes, NJ, USA), supplemented with 2.5 g/L mucin (Oxoid, Basingstoke, UK), 1.0 g/L yeast extract (Oxoid, Basingstoke, UK), 0.1 g/L cysteine (Sigma, St. Louis, MO, USA), 2.0 g/L sodium bicarbonate (Merck, Darmstadt, Germany), 5.0 mg/L haemin (Sigma, St. Louis, MO, USA), 1.0 mg/L menadione (Merck, Darmstadt, Germany), and 0.25% (*v*/*v*) glutamic acid (Sigma, St. Louis, MO, USA). After incubation, microbial growth was measured spectrophotometrically to develop a microbial suspension containing 10^6^ colony-forming units (CFU)/mL of each bacterium and, where appropriate, 10^4^ CFU/mL of *C. albicans*.

### 4.2. In Vitro Dynamic Multispecies Biofilm Model

An in vitro multispecies dynamic biofilm model was used [61,62], which has been validated on biofilms growing on implant surfaces [35,63]. Basically, the system consists of a sterile vessel where the liquid culture medium, namely the previously described protein-enriched BHI medium, is pumped into the bioreactor by a peristaltic pump at constant pressure. The bioreactor (Lambda Minifor^©^ bioreactor, LAMBDA Laboratory Instruments, Sihlbruggstrasse, Switzerland) maintains the culture medium under stable conditions at 37 °C, pH 7.2, and an anaerobic atmosphere (10% H_2_, 10% CO_2_, and N_2_ balance) during the whole incubation process. These conditions are maintained by directly pumping an anaerobic gas mixture (10% H_2_, 10% CO_2_, and equilibrium N_2_) through a filter into the incubation vessel, keeping the pressure constant. The system is inoculated with 5 mL of the previously described microbial suspension and maintained for 12 h under the described conditions. Subsequently, once the mixed culture reached the exponential growth phase, the continuous culture was activated through a second peristaltic pump with a flow rate of 30 mL/h to transfer the culture to Robbins devices placed in series that carry the sterile dental implant units on which the biofilm was developed (Straumann^®^ Tissue Level Standard, 8 mm in length and 3.3 mm in diameter, with the patented moderately rough sandblasted and acid-etched surface [Straumann Institute AG, Basel, Switzerland]). Inside the Robbins device, anaerobic conditions and a constant temperature (37 °C) are maintained during each experimental interval to allow biofilm development.

### 4.3. Experimental Groups

To evaluate the effect of *C. albicans* on the dynamics of subgingival *biofilm* formation on implant surfaces, three time intervals were analyzed: 24, 48, and 72 h. For each time interval, the developed biofilms were incubated under two different conditions, the test biofilms included a mixed culture composed of the bacterial strains *S. oralis*, *A. naeslundii*, *V. parvula*, *F. nucleatum*, *P. gingivalis*, *A. actinomycetemcomitans*, and the fungus *C. albicans*, while the control biofilms included only the six bacterial strains. At each time and in each condition, three implants were analyzed by confocal microscopy (CLSM) (*n* = 3), three by scanning electron microscopy (SEM) (*n* = 3), and nine by real-time polymerase chain reaction (qPCR) (*n* = 9).

### 4.4. Scanning Electron Microscopy (SEM)

After removal of the implants from the Robbins device, the implants were sequentially washed three times with 2 mL of phosphate-buffered saline (PBS) (immersion time per rinse, 10 s) to remove unattached bacteria. The implants were then fixed in a solution of 4% paraformaldehyde (Panreac Química, Barcelona, Spain) and 2.5% glutaraldehyde (Panreac Química) for 4 h at 4 °C. They were then washed in PBS and sterile water (immersion time per wash: 10 min) and dehydrated through a series of graded ethanol solutions (30%, 50%, 70%, 80%, 90%, and 100%; immersion time per series: 10 min). Then the specimens were dried, coated with gold, and analyzed using a JSM 6400 electron microscope (JSM6400, JEOL, Tokyo, Japan), with a backscatter electron detector and an image resolution of 25 kV. 

This analysis was carried out at the National Centre of Electron Microscopy (*Instalación Científico-Técnico singular*; ICTS) at the Moncloa Campus of the Complutense University of Madrid (Madrid, Spain).

### 4.5. Confocal Laser Scanning Microscopy (CSLM)

For the noninvasive confocal imaging of biofilms, a Leica LCS SP8 STED 3X microscope (Mannheim, Germany) was used. The CLSM software Leica Application Suite X version 3.5.7.23225 was configured to take a z-series of scans (XYZ) of 1 µm thickness (8 bits, 512 × 512 pixels).

Prior to the microscopic analysis, the Robbins device was taken from the bioreactor and carefully removed the implants, which were then washed three times with 2 mL of PBS (immersion time per rinse, 10 s) to remove unattached bacteria.

For observing and quantifying the biofilm bacteria, the samples were stained with the LIVE/DEAD^®^ BacLightTM bacterial viability kit solution (Molecular Probes, The Netherlands), which contains propidium iodide (PI) and SYTO9 nucleic acid dyes. With this method, dead cells or those with compromised viability are stained in red (PI), while cells with an intact membrane are stained in green (SYTO9). Implants were then coated with fluorochromes in a 1:1 ratio and incubated for 9 ± 1 min to obtain the optimal fluorescence signal at the corresponding wavelengths (SYTO9: 515–530 nm; PI: >600 nm). To observe and quantify *C. albicans,* implants were stained for 10 min with 3% Calcofluor White (CFW), thus obtaining an optimal signal using a wavelength of 405 nm. 

Representative implant surface locations involving both the peak of a thread and the bottom of the valley were selected for the CLSM analyses. 

The COMSTAT 2.1 software (www.comstat.dk) was used to calculate the biomass in micrometres^3^/micrometres^2^ (µm^3^/µm^2^) of the CLSM images.

The analysis was performed at the Biological Research Centre Margarita Salas (*Centro de Investigaciones Biológicas*, *Consejo Superior de Investigaciones Científicas*—CIB-CSIC), located at the Moncloa Campus of the Complutense University of Madrid (Madrid, Spain).

### 4.6. Quantitative Polymerase Chain Reaction (qPCR)

Prior to DNA isolation, the test and control implants were rinsed sequentially in 2 mL of sterile PBS three times (immersion time per rinse: 10 s) to remove unattached bacteria. To disaggregate the biofilms, implants immersed in 1 mL of sterile PBS were vortexed at maximum power at room temperature for 2 min. To exclude genetic material from nonviable cells, 100 µL of the obtained suspension was incubated with 100 µM PMA (propidium monoazide) prior to DNA extraction [53], and 100 µL of the same suspension was analyzed without the PMA treatment to calculate the viability percentages of each strain.

DNA was isolated using the commercial MolYsisComplete5 kit, Molzym (GmbH & CoKG, Bremen, Germany), according to the manufacturer’s instructions. Primers and probes were supplied by Life Technologies Invitrogen (Carlsbad, CA, USA), Applied Biosystems (Carlsbad, CA, USA), and Roche (Roche Diagnostic GmbH, Mannheim, Germany). 

The amplification reaction was performed in a total mix volume of 10 μL. Reaction mixtures contained 5 μL of Master Mix 2x (LC 480 Probes Master, Roche), optimal concentrations of primers and probes (900, 900, and 300 nM for *S. oralis*; 300, 300, and 300 nM for *A. naeslundii* and *P. gingivalis*; 750, 750, and 400 nM for *V. parvula*; 300, 300, and 200 nM for *A. actinomycetemcomitans*; and 600, 600, and 300 nM for *F. nucleatum*), and 2.5 μL of DNA extracted from the samples. The negative control was 2.5 μL of sterile water [nontemplate control (NTC)] (Roche). The primers and probes used were previously described [62]. 

The target used for the detection and quantification of the six selected bacterial species was the 16S rRNA gene of each of them. For the detection and quantification of *C. albicans*, an optimization process of qPCR targeting the ribosomal internal transcribed spacer (ITS) region was previously performed. The primers and probe designed by He et al. (forward: 5′-GGT GTT GAG GAG CAA TAC GAC-3′; reverse: 5′-AGA CCT AAG CCA TTG TCA-3′; probe: 5′-FAM-ATC CCG CCT TAC CAC TAC CG-TAMRA-3′) were used [64]. Primer concentrations of 600 and 600 nM and probe concentrations of 300 nM were set as optimal, which generated a standard curve whose equation was y = −3.3598x + 42.944, R^2^ = 0.9996, and a detection limit set at 10^2^ colony-forming units (CFU)/mL, with a 95% confidence interval. No cross-reaction with DNA from the bacterial strains used was observed.

The amplification program consisted of an initial cycle at 95 °C for 10 min, followed by 40 cycles at 95 °C for 15 s and 60 °C for 1 min. It was performed on a LightCycler^®^ 480 II thermal cycler (Roche Diagnostic GmbH, Mannheim, Germany). The microplates used for qPCR were LightCycler480 Multiwell-384 (Roche).

Each DNA sample was analyzed in duplicate. The quantification cycle (Cq) values were determined using the provided software (LC 480 Software 1.5, Roche). Quantification of cells was based on extrapolation with previously designed standard curves with the Cq values generated in qPCR vs. log CFU/mL. The correlation between Cq and CFU/mL values was automatically generated by the software (LC 480 Software 1.5, Roche).

### 4.7. Statistical Analysis

Quantitative data was expressed in colony-forming units per milliliter (CFU/mL) (qPCR) and bacterial biomass obtained by CLSM was expressed in µm^3^/µm^2^. Data were reported as means and standard deviations (SDs), and the Shapiro–Wilk goodness-of-fit tests were used to assess data normality. When the two sets of data compared showed a normal distribution, a T-test with Welch’s correction was applied. When at least one of the two groups did not show a normal distribution, a Mann–Whitney test was applied. Statistically significant differences were considered for *p*-values < 0.05. GraphPad Prism version 8.0.1 software was used for all data analysis.

## Figures and Tables

**Figure 1 ijms-25-03277-f001:**
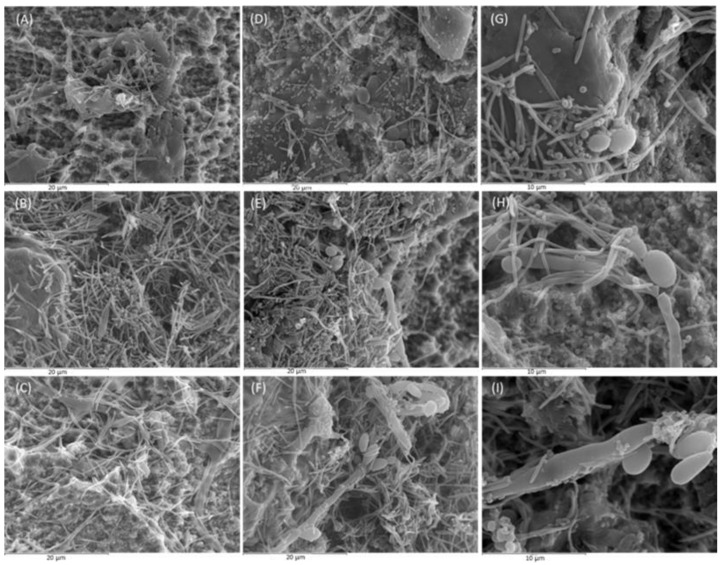
Images obtained by scanning electron microscopy (SEM) with 2500× magnification of control biofilms (in the absence of *Candida albicans*) developed at 24, 48, and 72 h ((**A**–**C**), respectively) and of test biofilms (in the presence of *C. albicans*) developed at the same intervals ((**D**–**F**), respectively). Images (**G**–**I**) show test biofilms with 5000× magnification after 24, 48, and 72 h of incubation, respectively (*n* = 6).

**Figure 2 ijms-25-03277-f002:**
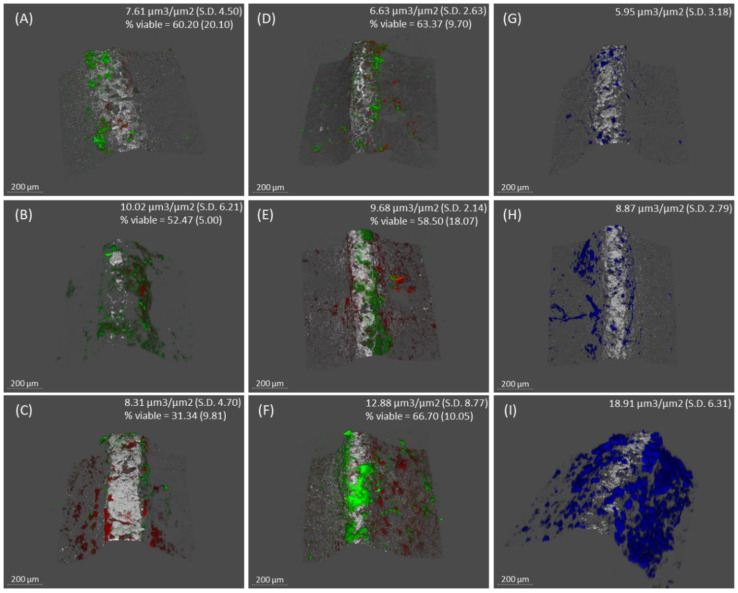
Images obtained by confocal laser scanning microscopy (CLSM) on control biofilms (in the absence of *Candida albicans*) developed at 24, 48, and 72 h ((**A**–**C**), respectively) and on test biofilms (in the presence of *C. albicans*) developed at the same intervals ((**D**–**F**), respectively). Images (**G**–**I**) show *C. albicans* at 24, 48, and 72 h, respectively (scale bar = 200 µm). LIVE/DEAD^®^ BackLight Kit was used to stain live bacteria (green), dead bacteria (red), and implant surfaces (white). Calcofluor White (CFW) was used to stain *C. albicans* (blue) (*n* = 6).

**Figure 3 ijms-25-03277-f003:**
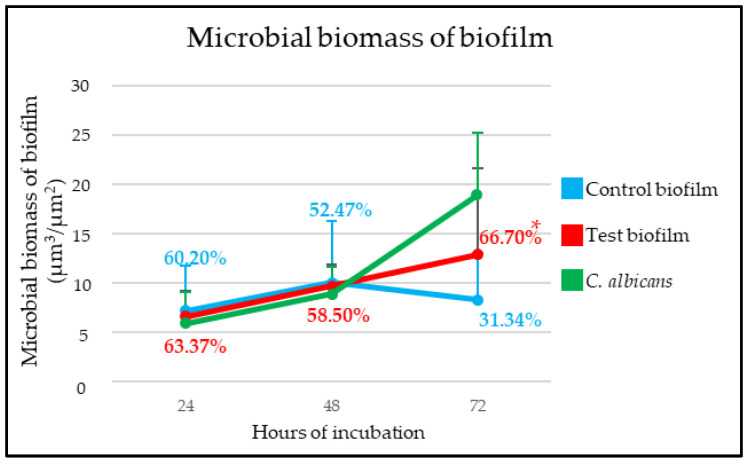
Kinetics of control and test biofilms and *Candida albicans* [expressed as microbial biomass of biofilm (µm^3^/µm^2^)] obtained by quantification of images of confocal laser scanning microscopy (CLSM). Percentages show the proportion of viable cells at each interval of incubation. * *p* < 0.05, statistically significant differences when comparing test and control biofilms at each time interval.

**Figure 4 ijms-25-03277-f004:**
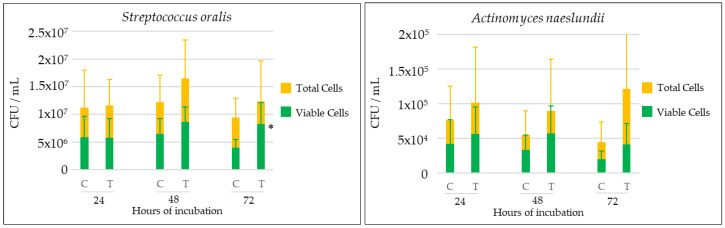
Kinetics (expressed as mean and standard deviation (SD)) of total and live microbial species (colony-forming units (CFUs)/mL) determined by quantitative polymerase chain reaction (qPCR) in 24, 48, and 72 h biofilms on dental implants in the presence (T, test) and absence (C, control) of *Candida albicans* (*n* = 9), using specific primers and probes directed to the 16S rRNA gene. * *p* < 0.05, statistically significant differences when comparing CFU/mL between test and control biofilms at each time interval. Comparisons between groups were performed considering viable cells and total cells. Figure corresponding to Appendix A.

**Figure 5 ijms-25-03277-f005:**
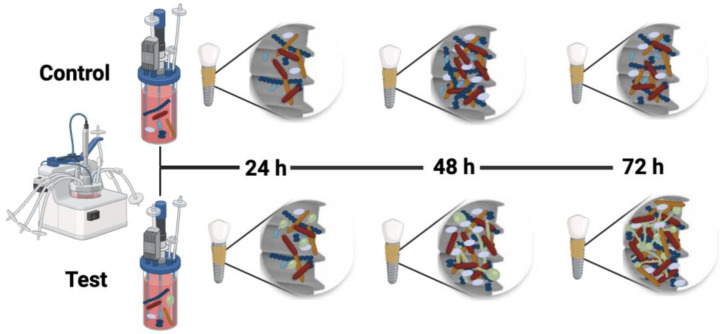
Graphical representation of the control biofilm, without *Candida albicans*, and the biofilm test developed in the presence of the fungus. Image was created on BioRender.com (accessed on 4 March 2024).

## Data Availability

Data are contained within the article.

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
