# Peer review of "The Impact of Candida albicans in the Development, Kinetics, Structure, and Cell Viability of Biofilms on Implant Surfaces—An In Vitro Study with a Validated Multispecies Biofilm Model"

_ijms, 2024, doi:10.3390/ijms25063277_

Round 1

Reviewer 1 Report

Comments and Suggestions for Authors

Title: The impact of Candida albicans in the development, kinetics, structure, and cell viability of biofilms on implant surfaces. An in vitro study with a validated multispecies biofilm model

The manuscript by Bravo et al. is interesting and requires minor revision as follows:

Comments:

1.      Introduction, the mechanism, and application of biofilm can be updated with their potential inhibitors.

2.      The author should provide one illustration based on the finding to present the mechanism and interactions for the desirable function.

3.      Table 1. Please verify the all data and it can be more precisely presented. The SD data are very high?

4.      Discussion could be improved with a few recent citations to justify the significance of this study.

Comments on the Quality of English Language

Moderate editing of English language is required.

Reviewer 2 Report

Comments and Suggestions for Authors

The manuscript describes the impact of Candida albicans in the development, kinetics, structure, and cell viability of biofilms on implant surfaces. This is a very important issue due to the increase in the use of dental implants and the consequent increase in peri-implant mucositis and peri-implantitis.

The study is relevant, with novelty, and well-structured. However, some changes should be made to publish the manuscript.

1)      Results section 2.1.: Microorganisms' names should be in italics. Please correct that in section 2.1

2)      Results section 2.1.: Since the materials and methods section is subsequent, it is necessary at the beginning of section 2.1 and in the caption of Figure 1 to specify what corresponds to the controls and tests.

3)      Results section 2.2.: It is necessary, in the caption of Figure 2, to specify what corresponds to the controls and tests.

4)      Results section 2.2.: In Figure 3 there is a clear decrease in C. albicans biomass at 72h. Something should be said about it.

5)      Discussion, Line 174, pag 7: Where it is “mature biofilms (72 h.) (Table 1). Similar”, should be “mature biofilms (72 h) (Table 1). Similar”.

6)      Discussion: In my opinion discussion should be reorganized. The authors start to discuss Table 1 results, then Figure 1, and then go back to Table 1. Also, the authors should discuss the results of C. albicans biofilm alone (Figure 3) and how is it related to other results.

7)      Materials and Methods section 4.1.: Microorganisms' names should be in italics. Please correct that in section 4.1

8)      Materials and Methods section 4.3.: C. albicans biofilm alone is not mentioned.

This is a valuable work that after these changes is publishable.

Reviewer 3 Report

Comments and Suggestions for Authors

This is a nice manuscript with good data on the interrelationship of oral microbial species. The text is well written and there are no major concerns about data and experiments. We believe however that the manuscript could significantly improve if the authors considered the following points: 

1. At the Introduction or Discussion, please explain why you analyzed the specific four bacterial species in the formation of biofilms assisted by the fungus Candida albicans. While you comment on the interrelation of some of these organisms (Discussion), you do not clearly state the significance/role of each examined bacterial species in oral pathogenesis. 

2. Table 1 is somewhat difficult to follow. Consider placing it to supplemental data and replace the table with bar graphs for each bacterial species. 

3. In the Materials and Methods, starting from line 260: use italics when referring to species and genera of organisms. Here and throughout all manuscript text. 

4. Line 361: it could be helpful to provide the concentration of template DNA used for the amplification reactions (line 365: 2.5 μL of DNA extracted from the samples.”. How much DNA was there?).

Round 2

Reviewer 1 Report

Comments and Suggestions for Authors

Accept 

Reviewer 2 Report

Comments and Suggestions for Authors

The authors have addressed all raised issues.

The manuscript has, in my opinion, quality and novelty to be published.